# Programmable Nanostructures Based on Framework-DNA for Applications in Biosensing

**DOI:** 10.3390/s23063313

**Published:** 2023-03-21

**Authors:** Bing Liu, Fan Wang, Jie Chao

**Affiliations:** 1School of Medicine & Holistic Integrative Medicine, Nanjing University of Chinese Medicine, 138 Xianlin Road, Nanjing 210023, China; 2State Key Laboratory of Organic Electronics and Information Displays & Jiangsu Key Laboratory for Biosensors, Institute of Advanced Materials (IAM), Jiangsu National Synergetic Innovation Center for Advanced Materials (SICAM), Nanjing University of Posts and Telecommunications, 9 Wenyuan Road, Nanjing 210023, China

**Keywords:** DNA nanostructures, biosensing, nucleic acids, virus, protein, bacteria

## Abstract

DNA has been actively utilized as bricks to construct exquisite nanostructures due to their unparalleled programmability. Particularly, nanostructures based on framework DNA (F-DNA) with controllable size, tailorable functionality, and precise addressability hold excellent promise for molecular biology studies and versatile tools for biosensor applications. In this review, we provide an overview of the current development of F-DNA-enabled biosensors. Firstly, we summarize the design and working principle of F-DNA-based nanodevices. Then, recent advances in their use in different kinds of target sensing with effectiveness have been exhibited. Finally, we envision potential perspectives on the future opportunities and challenges of biosensing platforms.

## 1. Introduction

In the last few decades, DNA has not only been a hallmark of molecular biology [1] but has also been used to assemble a variety of well-defined nanostructures. Several unique properties make DNA a robust material for nanoscale assembly and the keystone of nanodevices: (i) the Watson–Crick principle provides predictable self-assembly to associate different duplexes; (ii) compared to other biomolecules such as enzymes or antibodies, DNA is relatively stable in various biochemical environments [2]; (iii) the double helix has a diameter of about 2 nm and a helix spacing of about 3.5 nm, which makes it suitable for robust construction; (iv) synthetic DNA base sequences are inexpensive and readily available. Framework-like DNA nanostructures, dubbed “framework DNA” (F-DNAs), facilitate the different designs of DNA nanostructures with various features. First, there is a wide range of functionalized modification sites, which enable biomolecules or fluorescent dyes to be attached depending on the needs of the material [3]. Second, DNA nanostructures can be used as spatially addressable scaffolds at the nanoscale due to their sequence specificity at different sites. Third, compared to other nanomaterials, DNA nanomaterials are virtually non-cytotoxic and have good biocompatibility [4]. Today, many well-designed DNA nanostructures have been developed, and due to their apparent advantages, DNA nanostructures can be used for sensing [5,6,7,8], medication delivery [9,10], molecular transport [11,12], information processing [13], etc.

Biosensors hold tremendous potential for numerous applications such as clinical diagnosis, healthcare, and environmental monitoring [14,15]. With the advancements in medicine and increasing pursuit of health, the early diagnosis and prevention of diseases because of the ability to improve treatment outcomes and reduce treatment costs and mortality have elicited considerable interest. Biomarkers, including nucleic acids, proteins, microorganisms, and small molecules, play a significant role in disease detection and treatment follow-up [16]. The disease detection principle depends on the specific binding of the targets, such as antigen-antibody binding, nucleic acid aptamers that bind to particular targets, and similar inter-reactions. The needs for high specificity and sensitivity of biosensors make special recognition exceedingly important. Single-stranded DNA can be employed to identify target nucleic acids by a complementary sequence in diagnosing viral infections, bacterial infections, and cancer [17]. The modification of DNA contains several functional groups, such as a fluorophore, quencher pairs, and electrochemical redox labels to conjugate signaling moieties. In addition, a variety of additional reactive conjugates (for example, thiol groups for anchoring to a gold electrode surface or metal nanoparticles) make attachment tags relatively straightforward to conjugate different recognition elements on nucleic acid sequences, resulting in widely expanding the range of sensing targets of interest [18]. In fact, based on the precise addressability, F-DNA-based nanodevices with chemical versatility can be introduced, not limited to the targets usually recognized by DNA probes but also utilized as a versatile scaffold for a wide range of sensing applications [19].

In this review, we summarize the key developments in nanodevices based on F-DNA and provide a more distinct overview of biosensing approaches. We will not direct the review from the viewpoint of the wide range of nanodevices involving many materials: gold nanomaterials, quantum dots, graphene oxide, and so on; we will focus on the perspective of the types of disease analytes. This classification mode directs the readers to a well-defined and focused discussion on every analyte. The biosensor target examples based on F-DNA nanostructures we have included in this review can be divided into five major classes: nucleic acids, viruses, proteins, bacteria, and others. For each case, we provide the design principles of nanostructures and their features, including the read-out strategy and detection limit. Finally, we outline the future application prospects and development of the F-DNA direction.

## 2. Framework Nucleic Acid

Nucleic acids have been an attractive tool for constructing nanodevices. However, linear double helix structures do not meet the requirements for fabricating complex nanostructures. Branch DNA ligation (for example, the Holliday junction) is necessary for multidimensional assembly. Since Ned Seeman first devised the idea of using DNA to build structures in the early 1980s [20], two of the most notable contributions are the DNA tile and origami techniques. After decades of development, DNA tile self-assembly and origami-based self-assembly methods have exceeded the limits of self-assembly, leading to the development of a range of highly variable and complexly structured DNA devices.

### 2.1. The Design of the Stiffer Motif

The early artificial DNA structure is a stick-cube whose edges are double helices [21]. Other polyhedral forms, such as the Borromean ring and a truncated octahedron, have more complex topologies [22]. The basic principle is that, based on the shape you want, different strands of DNA are designed to bind to settled regions with complementary strands. Due to the apparent floppiness of individual branched junctions, a significant challenge of DNA nanotechnology is to generate more rigid connections, which are essential to achieving well-defined 2D DNA assemblies [23]. In subsequent studies, a series of rigid structural motifs were successfully designed, leading to the rapid development of structural DNA nanotechnology [23]. Such as the Holliday junctions, Seeman et al. combine two duplexes with a single strand of DNA from each crossover, called a crossover junction [24]. Two such crossover junctions can be connected to form two crosses, and the synthesized structural tile is called a double-crossover (DX) tile [25], as shown on the left of Figure 1a. Then, by adding a third domain, exchanges occur again between chains of opposite polarity, called the triple-crossover (TX) [26] motif, as shown in the middle of Figure 1a. The paranemic-crossover (PX) motif [27] is formed by exchanges between strands of identical polarity at every possible position, as shown in the right of Figure 1a. They have been used to construct many DNA lattices, such as cross-shaped double-decker tiles [28], tensegrity triangles [29], three-point star motifs, and six-point star motifs [30,31], as illustrated in Figure 1b–e. Subsequently, using a single-step annealing method, a DNA tetrahedron consisting of four strands is synthesized [32], in which the triangular structure gives stability to the design.

DNA nanodevices are expected to develop complicated shapes in three dimensions to enable better applications in the biomedical field, such as drug delivery and structure size programmability. A large number of 3D nanostructures can be constructed directly from 2D DNA shapes, providing a simple method for this desire [33], as shown in Figure 1f. For example, DNA motifs of the three-point star are widely used to synthesize cubes, nanocages, and symmetric supramolecular polyhedral [30,34,35,36,37], as demonstrated in Figure 1g,h. Similarly, a DNA icosahedron can be assembled from five-point star DNA motifs [38], which has 20 triangular faces and 12 five-branched vertices, demonstrating the flexibility of DNA nanostructures. The first rationally designed 3D DNA crystal was created by using the tensegrity triangle [39]. A two-turn tensegrity triangle motif connects to six others via sticky-end cohesion. The cavities in the designed crystal can be used to attach and hold other macromolecules and for crystallographic analysis, which brings us one step closer to realizing the original vision of DNA nanotechnology.

### 2.2. DNA Origami

Based on the tile self-assembly, a single-stranded DNA molecule of 1669 nucleotides is folded into an octahedral structure by a simple denaturation-complexity process in the presence of 5 40-mer synthesized oligodeoxynucleotides [40], which later became the forerunner of the iconic event DNA origami. In 2006, DNA nanotechnology produced a breakthrough when Paul Rothemund invented DNA-based origami technology [41]. Since then, more self-assembling devices based on origami DNA have been designed. In a sense, DNA origami can be seen as a large composite of DX motifs. A long single-stranded DNA called a scaffold, usually a viral genome, and multiple small “short peg” strands are used to fold the scaffold strands into a pre-designed shape.

Using this design strategy, other groups have successfully constructed 2D planar patterns, such as a portrait of dolphins and a ribbon lattice [42,43], as shown in Figure 1i. Han et al. proposed a strategy to create curvature within 2D planar structures [44], which has been successfully implemented to engineer DNA into concentric rings. Furthermore, many single-layered designs of 3D shapes had been created. For example, by routing a scaffold through four triangular faces with single-stranded hinges, Ke et al. created a tetrahedral structure [45], as illustrated in Figure 1j. Other groups used a stepwise folding mechanism controlled by staples to assemble a cube box and a DNA prism [46,47], as shown in Figure 1k. In combination with an out-of-plane curvature strategy, complex structures, such as nanosized flasks, can also be fabricated [44]. New structures can be achieved if new patterns are introduced into the origami structure. Utilizing multiarm junctions, Han et al. were the first to create a grid pattern for wireframe DNA origami [48]. Curved designs such as a sphere can be created by applying the bending DX structures with tension to the gridiron units. Complex wireframe patterns, such as images of flowers, birds [49], and rabbits [50], were created by joining together dobby arms at many different angles. Yin et al. have developed a more straightforward “short single-stranded tile” (SST)-based approach to demonstrate a large number of nanodevices based on micro-origami designs [51]. Unlike conventional DNA tiles, SST tiles are small, single-stranded, and have no ordered structure before being incorporated into the superstructure. This design is similar to DNA-based Lego blocks [52]. In this strategy, short sequences of DNA are used as tiles that self-assemble with identical tiles.

**Figure 1 sensors-23-03313-f001:**
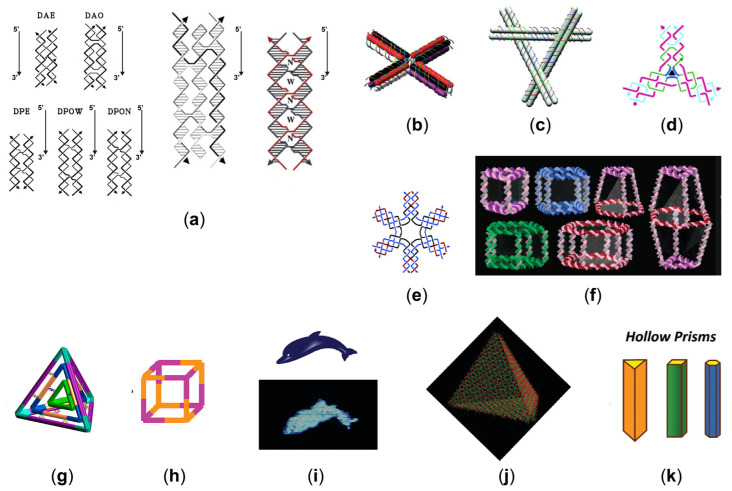
(**a**) The left is the five models of DNA double crossover molecules [25]. (Reprinted from *Biochemistry*, T. J. Fu, and N. C. Seeman, DNA double-crossover molecules, 3211–3220, Copyright (1993), with permission from the American Chemical Society). The middle one is TX [26], which contains three helices and four strands. (Reprinted from the *Journal of the American Chemical Society*, LaBean, T.H.; Yan, H.; Kopatsch, J.; Liu, F.; Winfree, E.; Reif, J.H.; Seeman, N.C. Construction, Analysis, Ligation, and Self-Assembly of DNA Triple Crossover Complexes, **2000**, *122*, 1848–1860, Copyright (2000), with permission from the American Chemical Society). The last one is PX [27], which contains four strands, arranged in two double-helical domains related by a central dyad axis. (Reprinted from the *Journal of the American Chemical Society,* Shen, Z.; Yan, H.; Wang, T.; Seeman, N.C. Paranemic Crossover DNA: A Generalized Holliday Structure with Applications in Nanotechnology, **2004**, *126*, 1666–1674, Copyright (2004), with permission from the American Chemical Society). (**b**) Cross-shaped double-decker tile [28]. (Reprinted from the *Journal of the American Chemical Society*, Majumder, U.; Rangnekar, A.; Gothelf, K.V.; Reif, J.H.; LaBean, T.H. Design and construction of double-decker tile as a route to three-dimensional periodic assembly of DNA, **2011**, *133*, 3843–3845, Copyright (2011), with permission from the American Chemical Society). (**c**) Tensegrity triangle with double-crossover edges [29]. (Reprinted from *Nano Letters*, Zheng, J.; Constantinou, P.E.; Micheel, C.; Alivisatos, A.P.; Kiehl, R.A.; Seeman, N.C. Two-Dimensional Nanoparticle Arrays Show the Organizational Power of Robust DNA Motifs, **2006**, *6*, 1502–1504, Copyright (2006), with permission from the American Chemical Society). (**d**) Three-point star motif [30]. (Reprinted from the *Journal of the American Chemical Society*, He, Y.; Chen, Y.; Liu, H.; Ribbe, A.E.; Mao, C. Self-Assembly of Hexagonal DNA Two-Dimensional (2D) Arrays, **2005**, *127*, 12202–12203, with permission from the American Chemical Society). (**e**) Six-point star motif [31]. (Reprinted from the *Journal of the American Chemical Society*, He, Y.; Tian, Y.; Ribbe, A.E.; Mao, C. Highly Connected Two-Dimensional Crystals of DNA Six-Point-Stars, **2006**, *128*, 15978–15979, Copyright (2006), with permission from the American Chemical Society). (**f**) A large number of three-dimensional discrete DNA assemblies [33]. (Reprinted from the *Journal of the American Chemical Society*, Aldaye, F.A.; Sleiman, H.F. Modular Access to Structurally Switchable 3D Discrete DNA Assemblies, **2007**, *129*, 13376–13377, Copyright (2007), with permission from the American Chemical Society). (**g**) Self-assembly of a multilayered DNA tetrahedron [35]. (Reprinted from the *Journal of the American Chemical Society*, Liu, Z.; Tian, C.; Yu, J.; Li, Y.; Jiang, W.; Mao, C. Self-assembly of responsive multilayered DNA nanocages, **2015**, *137*, 1730–1733, Copyright (2015), with permission from the American Chemical Society). (**h**) DNA tubes of self-assembly from eight copies of identical three-point star tiles [36]. (Reprinted from the *Journal of the American Chemical Society,* Zhang, C.; Ko, S.H.; Su, M.; Leng, Y.; Ribbe, A.E.; Jiang, W.; Mao, C. Symmetry Controls the Face Geometry of DNA Polyhedra, **2009**, *131*, 1413–1415, Copyright (2009), with permission from the American Chemical Society). (**i**) The origami of dolphin shape [42]. (Reprinted from *ACS Nano*, Andersen, E.S.; Dong, M.; Nielsen, M.M.; Jahn, K.; Lind-Thomsen, A.; Mamdouh, W.; Gothelf, K.V.; Besenbacher, F.; Kjems, J. DNA Origami Design of Dolphin-Shaped Structures with Flexible Tails, **2008**, *2*, 1213–1218, Copyright (2008), with permission from the American Chemical Society). (**j**) A DNA tetrahedron [45]. (Reprinted from *Nano Letters*, Ke, Y.; Sharma, J.; Liu, M.; Jahn, K.; Liu, Y.; Yan, H. Scaffolded DNA Origami of a DNA Tetrahedron Molecular Container, **2009**, *9*, 2445–2447, Copyright (2009), with permission from the American Chemical Society). (**k**) Hollow prism of folding multiple-arm DNA structures [47]. (Reprinted from the *Journal of the American Chemical Society,* Endo, M.; Hidaka, K.; Kato, T.; Namba, K.; Sugiyama, H. DNA Prism Structures Constructed by Folding of Multiple Rectangular Arms, **2009**, *131*, 15570–15571, Copyright (2009), with permission from the American Chemical Society).

## 3. Applications of DNA Nanostructures for Biosensing

### 3.1. Applications of DNA Nanotechnology in Nucleic Acids

Nucleic acid analysis techniques are widely used in laboratory research and clinical diagnostics. To date, many effective strategies have been developed for sensitive and specific detection at low cost.

In these strategies, oligonucleotides are often used as molecular recognition elements. Watson–Crick bases as pairing rules and sensing events are made highly sensitive and tunable by the rational design of probe sequences [53,54]. Other nanomaterials, represented by metal and magnetic nanoparticles, are also widely used in nucleic acid sensor research due to their unique optoelectronic properties, size effect, surface effect, and chemical stability [55].

#### 3.1.1. DNA

For improving the intensity of biomolecular signal transduction in various settings, a chemically cross-linked branched DNA nanostructure (CCLB-DNA) is designed as a probe DNA to construct biological interfaces for ultra-sensitive nucleic acid detection [56], in which DNA-functionalized Fe_3_O_4_ nanoparticles are used for signal amplification. For multiplex ctDNA identification, a three-dimensional coding chain DNA loop (3D coding ID loop) platform was created [57]. The specific binding complex of ctDNA to the recognition loop initiates target-responsive cleavage by restriction endonucleases, triggering amplification of the rolling circle on the reporter loop. The sensitivity of this method is much higher than that obtained by sequencing. In addition to static DNA structures, dynamic DNA nanotechnology enables the manipulation and realization of corresponding functions through controlled motion or reconstruction [58]. DNA nanotweezers (DTs) are reversible DNA nanodevices that can optionally be switched between the on and off states involving the structure change. A regenerative DNA tweezer was designed to dynamically regulate the inter-enzyme spacing for efficient enzyme cascade amplification for homogeneous determination of target DNA associated with cancer, with the dynamic and reversible regulation of enzyme cascade catalytic efficiency [59], as shown in Figure 2a. Subsequently, using targeted switching DNA nanotweezers as a hemin concentration controller and combining a G-quadruplex structure in which the rich-G bases of the different strands are stabilized together by hydrogen bonding to form a tetrameric structure, an ultrasensitive “on-off” ECL biosensor was developed for the detection of PML/RARα, the important biomarker for acute promyelocytic leukemia (APL) [60]. Furthermore, DNA origami combined with the nanopores can also be designed to detect target DNA [61]. Nanopores in DNA origami-graphene heterostructures produce distinguishable residence times and ionic currents for four DNA base types for DNA detection.

#### 3.1.2. MicroRNA

MicroRNAs (miRNAs) are a class of endogenous, small, single-stranded non-coding RNAs that play an important role in many life events by regulating gene expression. The expression profile of miRNAs may be altered in the development of some diseases, such as cancer and autoimmune diseases. Therefore, they are expected to be diagnostic or prognostic biomarkers in the safety evaluation of drugs. A Hemin/G-quadruplex complex, often called a peroxidase-mimicking DNAzyme, can be well-designed to detect the various biomarkers by electrochemistry [62]. Additionally, G-quadruplex as a signal transduction device can achieve sensitive detection of miRNA-21 by colorimetry [63], which has detection limits as low as 1 aM of miR-21 and provides an excellent capability to discriminate single-base mismatches. A highly sensitive method for detecting miRNA-21 in breast cancer is the hairpin-mediated secondary enzyme amplification detection strategy [64], which is ten fM of the detection limit at 37 °C and 1 aM at 4 °C. More importantly, the method is sensitive and selective when applied to crude extracts of MCF-7 and PC3 cell lines and even to tissue from patients with intraductal and invasive ductal carcinoma of the breast. Hybridization chain reaction (HCR), a toehold-mediated strand displacement (TMSD) reaction, has attracted great interest because of its enzyme-free nature, isothermal conditions, simple protocols, and excellent amplification efficiency. Recently, a one-pot cascade signal amplification strategy based on an integrated CRISPR/Cas13a system (Cas13a) and a branched hybridization chain reaction (bHCR) was developed with the excellent surface-enhanced Raman scattering (SERS) properties of silver nanorods [65], also shown in Figure 2b. For the detection of gastric cancer-associated miR-106a, the procedure can be completed in less than 60 min with a limit of detection (LOD) as low as 8.55 aM. DNA tetrahedral nanostructures have been widely studied and applied in microbial identification, medical diagnostics, and biosensors due to their good biocompatibility, excellent cell membrane permeability, simple preparation, high yield, and adjustable size and dynamics [32]. For example, in order to improve the analytical sensitivity in the quantification of miRNAs, a DNA tetrahedron (DT) functionalized MB was designed and further coupled to substrate-modified AuNPs (Sub-AuNP). Based on that, the MB-DT multicomponent nucleic acid enzyme (MNAzyme) system with AuNPs as elemental markers was proposed for the highly sensitive quantification of miRNA-155 by ICP-MS [66]. A new method for the simultaneous detection of miRNA-21, miRNA-122, and miRNA-223 was presented based on DNA tetrahedral nanolabeling and fluorescence resonance energy transfer (FRET) [67]. The FRET effect between TOTO-1 and the other three fluorescent dyes has been shown to allow multiple sensitive quantifications of the three miRNAs in 10% of human serum samples. In order to enable a nanodevice with a self-penetrating capability in cells, a self-assembly strategy based on Y-DNA hybridization consisting of three nucleotide chains and chiral AuNP bilayer nuclear satellite structures with high region-controllability were constructed to detect miRNAs in living cells [68]. Furthermore, detecting miRNAs directly from exosomes could be a promising method for early disease diagnosis and therapeutic efficacy assessment. Traditional assays include exosome isolation [69] and reverse transcription-quantitative polymerase chain reaction (RT-qPCR) analysis, which is often expensive, laborious, and experimentally challenging [70,71]. A DNA zipper-mediated membrane fusion method for rapid detection of exosome miRNAs can effectively solve the challenges of high cost and time consumption in traditional exosome miRNA detection [72]. Different from the above detection in homogeneous solutions, a highly ordered 3D DNA nanomachine assembled from an azobenzene-functionalized DNA clamp has been developed in solid-state sensors. The initial state of the nanomachine is a “closed” conformation in which the electrochemiluminescence (ECL) emitter Ru(bpy)_2_^2+^ and the burster Alexa Fluor (AF) are separated, showing a “signal off” ECL response. After the miRNA interacts with the nanomachine, the clamp is “turned on” to hybridize with the miRNA, and the proximity between Ru(bpy)_2_^2+^ and AF leads to an enhanced ECL signal. This 3D DNA nanomachine has an organized and highly localized clamp that exhibits higher efficiency than conventional Au-based 3D nanomachines, resulting in the sensitive detection of miRNA-21 (LOD: 6.6 fM).

#### 3.1.3. mRNA and Circle RNA

mRNA is a template that directly guides protein biosynthesis and is involved in information transmission in gene expression. Circular RNAs (circRNAs) are a class of non-coding RNA molecules that form a circular structure with covalent bonds. Importantly, mRNA can be used as a biomarker for cancer therapy [73], and circRNAs resistant to enzyme degradation are aberrantly expressed in cancer and play an integral oncogenic or anticancer role in tumor development [74].

A molecular beacon (MB) is highly sensitive and selective in biosensing technology. MBs with stem-loop structures undergo fluorescence production and quenching depending on the presence or absence of the target. Moreover, MBs offer thermodynamic stability, the option to use different fluorophore combinations, excellent sensitivity, and real-time detection [75]. An aptamer-based MB was created for intracellular mRNA analysis [76], which was modified chemically and structurally to provide light-controlled and targeted delivery. In addition to MB, enzyme-free isothermal signal amplification has the advantages of high specificity, high sensitivity, good reproducibility, simple and convenient operation, and easy automation, which allows the target signal to be cyclically amplified for detection [77,78]. HCR is an important technique that is used in many biosensing platforms. Wu et al. developed an electrostatically assembled nucleic acid nanostructure, proving HCR in living cells to achieve sensitive detection of intracellular mRNA [79].

Rolling circle amplification (RCA) is a simple and highly efficient isothermal enzymatic amplification strategy to synthesize ultralong single-stranded DNA. With a good combination of RT-RCA [80], the two-dimensional reticulated HCR consists of a cleverly designed trigger strand and two hairpin fuel probes. A stable network structure can be generated at a constant temperature with RT-RCA products containing multiple sets of repeats, resulting in an enhanced fluorescent signal. Good selectivity allows the detection of circRNAs down to 0.1 pM. Catalytic hairpin assembly reaction (CHA) is a novel isothermal amplification technique for nucleic acid probe signal amplification proposed by Yin et al. in 2008 [81], with a lower background signal and a more stable reaction system. Based on that, an alternative significant signal amplification method and chameleon DNA template silver nanoclusters (DNA-AgNCs) were used for label-free ratio detection of circRNA [82]. As illustrated in Figure 2c, upstream CHA1 can be triggered explicitly by the analyte to form a double-stranded DNA (dsDNA) product, reducing the red fluorescence of hairpin DNA-AgNCs due to structural disruption. The leaky trigger sequence can further activate downstream CHA2 to generate another dsDNA complex, inducing other dark AgNCs to approach the G-rich sequence and resulting in an intense increase in green fluorescence. Thus, measuring the apparent change in the green-to-red fluorescence intensity ratio can be used for sensitive detection and visual differentiation of circRNA. The fluorescence signal holds the intrinsic drawback of a strong background signal sourced from external influence. In order to circumvent these issues, carbon fiber microelectrodes (CFME) were introduced. By targeting the BSJ site of circRNA, a peptide nucleic acid (PNA) probe can detect circRNA [83]. As exemplified in Figure 2d, Au nanoflowers (AuNFs) carrying PNA probes were modified onto CFME. The lower DPV signal is due to the stronger electrostatic repulsion of the circRNAs sequence towards the ferricyanide [Fe (CN)_6_]^3−^ ion. This results in an electrochemical point-of-care testing POCT platform with a linear range of 10 fM–1 μM and a detection limit as low as 3.29 fM.

**Figure 2 sensors-23-03313-f002:**
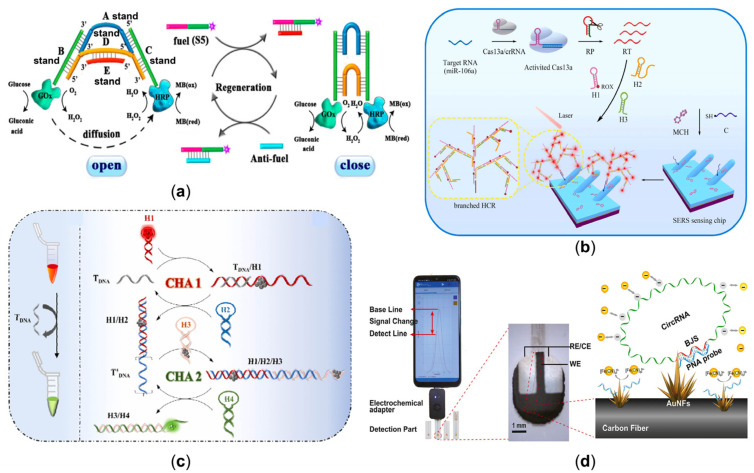
(**a**) Electrochemical DNA biosensor based on Mg^2+^-dependent DNAzyme cleavage recycling amplification and a regenerated DNA tweezer’s dynamical regulation of the enzyme cascade reaction [59]. (Reprinted from *Analytical Chemistry*, Kou, B.; Chai, Y.; Yuan, Y.; Yuan, R. Dynamical Regulation of Enzyme Cascade Amplification by a Regenerated DNA Nanotweezer for Ultrasensitive Electrochemical DNA Detection, **2018**, *90*, 10701–10706, Copyright (2018), with permission from the American Chemical Society). (**b**) The sensing of Cas13a-bHCR utilizing cascade signal amplification for detection of miRNA [65]. (Reprinted from *Biosensors and Bioelectronics*, Zhang, J.; Song, C.; Zhu, Y.; Gan, H.; Fang, X.; Peng, Q.; Xiong, J.; Dong, C.; Han, C.; Wang, L. A novel cascade signal amplification strategy integrating CRISPR/Cas13a and branched hybridization chain reaction for ultra-sensitive and specific SERS detection of disease-related nucleic acids, **2023**, *219*, Copyright (2023), with permission from Elsevier). (**c**) Hairpin-mediated fluorescence changes triggered by the target circRNA [82]. (Reprinted from *Biosensors and Bioelectronics*, Yang, M.; Li, H.; Li, X.; Huang, K.; Xu, W.; Zhu, L. Catalytic hairpin self-assembly regulated chameleon silver nanoclusters for the ratiometric detection of CircRNA, **2022**, *209*, 114258, Copyright (2022), with permission from Elsevier). (**d**) The process of electrochemical detection based on AuNFs-CFME [83]. (Reprinted from *Biosensors and Bioelectronics*, Zhang, B.; Chen, M.; Cao, J.; Liang, Y.; Tu, T.; Hu, J.; Li, T.; Cai, Y.; Li, S.; Liu, B.; et al. An integrated electrochemical POCT platform for ultrasensitive circRNA detection towards hepatocellular carcinoma diagnosis, **2021**, *192*, 113500, Copyright (2021), with permission from Elsevier).

### 3.2. Applications of DNA Nanotechnology in Virus

#### 3.2.1. Virus Nucleic Acid Detection

The virus can be diagnosed early by testing for viral nucleic acids. The polymerase chain reaction (PCR) as an efficient nucleic acid amplification technique has been considered the gold standard for low-abundant HIV DNA detection [84]. Combining with PCR and hemin/G-quadruplex, a multi-signal synergistic amplification method for the electrochemical sensing of HIV DNA was proposed based on a 2D DNA-Au nanowire structure. The signal amplification element includes hemin/G-quadruplex and a 2D DNA-Au nanowire structure, which enabled ultra-sensitive detection of HIV DNA with a detection limit of 1.3 aM. As an alternative to conventional methods, surface plasmon resonance (SPR) biosensing is a powerful analytical tool due to its apparent advantages of regular, label-free, real-time monitoring and analysis [85]. Based on that, Diao et al. proposed the high-sensitivity detection of HIV-associated DNA using entropy-driven strand displacement SPR reactions (ESDRs) and double-layer DNA tetrahedra (DDTs) [86]. The target DNA can explicitly trigger ESDRs to form an abundant double-stranded DNA (dsDNA) product. Then, they can bind to the immobilized hairpin capture probe and attach to the DDT’s nanostructure. The SPR response signal was significantly enhanced due to the high efficiency of ESDRs and the significant molecular weight of DDTs, which achieved a sensitive and specific detection of target DNA over a linear range of 1 pM–150 nM with a detection limit of 48 fM.

COVID-19 has caused a rapidly evolving pandemic. Current assays rely primarily on target amplification, which requires extensive replication of target sequences before detection. RT-qPCR is, therefore, widely used. However, it requires thermal cycling, prolonged target amplification processing, and a lack of programmability to adapt to new pathogen variants. So, there is a great need to develop a robust, fast, convenient, affordable, and ultra-sensitive assay/kit for detecting SARS-CoV-2. For example, integrated molecular nanotechnology was constructed for direct and programmable detection of SARS-CoV-2 RNA targets with a coupling of responsive equilibrium in an enzymatic network [87]. In further research, the automation level of detection can be improved by introducing automated microfluidics technology. As exemplified in Figure 3a, combined with amplifying electrochemistry, the design of an electrochemical system integrating reconfigurable enzyme-DNA nanostructures (eSIREN) utilizing embedded electronics can improve analytical performance and automation [88]. A DNA molecular complex containing a polymerase was designed, in which the enzyme activity was inhibited. When the target RNA is present, the polymerase activity is released, extending the signaling nanostructure to incorporate biotin-modified deoxynucleotide triphosphates (biotin-dNTPs), thereby enriching HRP and enhancing the electrochemical signal near the electrode surface. As the new coronavirus continues to evolve, different strains of pathogenic viruses are derived. To detect viral RNA variants, Zhang et al. used thermodynamic energy penalties associated with nucleic acid strand substitution reactions, single base pair mismatches, and metal ion-controlled urease cleavage to amplify the recognition of viral RNA [89]. A smartphone was outfitted to read out changes in pH by colorimetric methods, which are customizable and inexpensive. Since respiratory RNA levels rapidly decline after infection, plasma SARS-CoV-2 RNA may represent a viable alternative diagnostic [90]. Ning et al. reported an assay for the SARS-CoV-2 gene targets by fusing extracellular vesicles captured directly from plasma with reagent-loaded liposomes [91], providing a tool to accurately identify patients, including the cases without SARS-CoV-2 RNA detectable in the respiratory tract.

#### 3.2.2. Virus Protein Detection

Antigen-antibody reactions, such as the specific interaction with glycoproteins on the surface of the virus, are a mainstay for its detection. However, these immunoassay methods are time-consuming and can directly result in virus transmission and patient deterioration. Unlike antibodies, aptamers—short strands of ssDNA or RNA binding to specific target molecules or substances—are robust, easily chemically synthesized, and modified, which has been utilized to recognize the differing protein morphologies of different viruses. As illustrated in Figure 3b, a simple electrochemical aptamer sensor for detecting SARS-CoV-2 S1 protein had been established using the dumbbell hybridization chain reaction (DHCR) technique [92]. The aptamer sequence of the SARS-CoV-2 S1 protein was previously positioned by partial hybridization with the DHP0 of the triangular prismatic DNA (TPDNA) nanostructure. After incubation with the target protein, the aptamer sequence was displaced, and DHP0 was exposed. By introducing the DHCR fuel frame, the compressed DNA linearly assembles and enriches the electrochemical component at the electrode interface. Viral aptamers can be attached to scaffolds to improve polyvalent binding affinity.

However, some viruses, such as the dengue virus (DENV), have complex geometric patterns. Existing stents cannot address them because they are imprecise in ligand spacing or have limited control over stent shape and ligands. Therefore, a star-shaped DNA structure for biosensing was conceived, which was coupled with ten dengue envelope protein structural domain III (ED3) targeting aptamers in a two-dimensional pattern that can precisely match the ED3 cluster on the surface of the dengue (DENV) virus [93]. As the DENV envelope proteins are rigid, they are arranged in a shell-like structure on the virus’s surface, making them suitable for star-rigid matching. Furthermore, not all viruses have wooden frames. For example, the envelope glycoproteins of the new coronavirus have flexible stems and mobile roots, so a single two-dimensional model is no longer adequate. As shown in Figure 3c, a reticulated DNA nanostructured strategy was designed to selectively identify and capture intact SARS-CoV-2 viral particles with high affinity through spatial pattern matching and multivalent interactions between aptamers located on the DNA web and trimeric spike glycoproteins displayed on the outer surface of the virus [94]. The design highlights that when exposed to SARS-CoV-2 virus particles, the DNA net can bend itself to match the trimeric proteins on the surface of the virus particles. Using a convenient fluorescent agent, a fluorescent signal can be detected when the virus binds to DNA net-aptamers carrying the designed nano-switch, which can also be read in a laboratory setting using a high-throughput platform. To overcome the limitation of the ability of monomeric aptamers to bind only one subunit of the spike protein. As shown in Figure 3d, a triple rotationally symmetrical branching homo-trimeric aptamer has been developed for the symmetrical shape of the SARS-CoV-2 stinger protein [95]. This perfectly complementary structural scaffold greatly enhanced binding affinity. As research progresses, more and more ideas will be developed to detect novel coronavirus mutant strains. Yang et al. identified a SARS-CoV-2 Omicron variant RBD-binding adaptor (SCORe) in the SELEX process using the BA.1S1 protein [96], which was essential for the rapid detection of SARS-CoV-2 in patient samples and for the emerging technology of sensing intact viruses in the environment.

**Figure 3 sensors-23-03313-f003:**
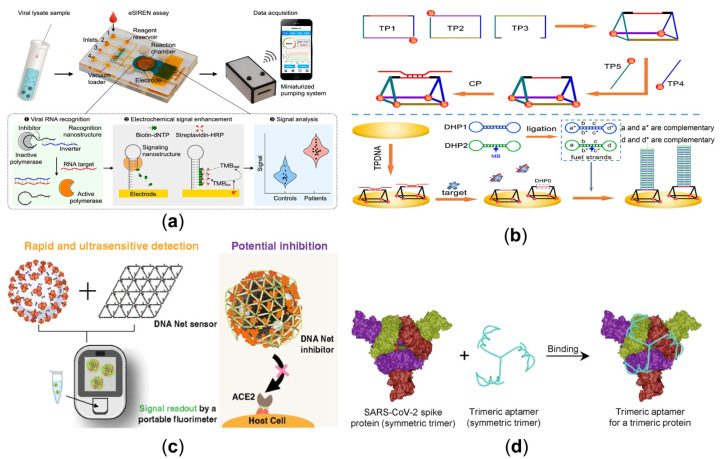
(**a**) Molecular nanostructures with electrochemical signal enhancement to detect SARS-CoV-2 by automated microfluidics [88]. (Reprinted from *Biosensors and Bioelectronics*, Zhao, H.; Zhang, Y.; Chen, Y.; Ho, N.R.Y.; Sundah, N.R.; Natalia, A.; Liu, Y.; Miow, Q.H.; Wang, Y.; Tambyah, P.A.; et al. Accessible detection of SARS-CoV-2 through molecular nanostructures and automated microfluidics, **2021**, *194*, 113629, Copyright (2021), with permission from Elsevier). (**b**) Aptamer-mediated target recognition with the DHCR process for the detection of the virus [92]. (Reprinted from *Analytical Chemistry*, Jiang, Y.; Chen, X.; Feng, N.; Miao, P. Electrochemical Aptasensing of SARS-CoV-2 Based on Triangular Prism DNA Nanostructures and Dumbbell Hybridization Chain Reaction, **2022**, *94*, 14755–14760, Copyright (2022), with permission from the American Chemical Society). (**c**) Using the DNA Net-aptamer for the viral capture and reading/inhibition [94]. (Reprinted from the *Journal of the American Chemical Society,* Chauhan, N.; Xiong, Y.; Ren, S.; Dwivedy, A.; Magazine, N.; Zhou, L.; Jin, X.; Zhang, T.; Cunningham, B.T.; Yao, S.; et al. Net-Shaped DNA Nanostructures Designed for Rapid/Sensitive Detection and Potential Inhibition of the SARS-CoV-2 Virus, **2022**, Copyright (2022), with permission from the American Chemical Society). (**d**) Complementarity of SARS-CoV-2 trimer spike proteins with trimer aptamers for molecular recognition [95]. (Reprinted from the *Journal of the American Chemical Society,* Li, J.; Zhang, Z.; Gu, J.; Amini, R.; Mansfield, A.G.; Xia, J.; White, D.; Stacey, H.D.; Ang, J.C.; Panesar, G.; et al. Three on Three: Universal and High-Affinity Molecular Recognition of the Symmetric Homotrimeric Spike Protein of SARS-CoV-2 with a Symmetric Homotrimeric Aptamer, **2022**, *144*, 23465–23473, Copyright (2022), with permission from the American Chemical Society).

### 3.3. Applications of DNA Nanotechnology in Protein

Many diseases are implicated with misfolded, truncated, mutated, and over- or under-expressed proteins, making their detection necessary for clinical diagnosis. DNA can be used as a probe for protein detection. A nucleic acid nano-switch platform had been developed to allow the transient measurement of G and E immunoglobulin (IgG and IgE) levels directly in serum and other body fluids [97]. Subsequently, to make cell-free nucleic acid diagnostics applicable to a wide range of targets, a cell-free in vitro transcription system using an antigen-coupled DNA conformational switch was reported to detect specific target antibodies [98]. Due to incomplete promoter sequence formation, the dsDNA switch adopted a stem-loop hairpin conformation that prevented efficient RNA ignition aptamer transcription. When the target antibody is bound to the two antigen-coupled DNA strands, hybridization forms a functional bimolecular complex. This complex induced a conformational change on the switch, restoring the promoter sequence to its integrity. In the presence of RNA polymerase and nucleotides, the activated transcription switch transcribed the lit RNA aptamer reporter, indicating the presence of the target antibody. In addition, developing detection methods for tumor makers with improved sensitivity and simplified reaction steps is still significant. The isothermal nucleic acid amplification techniques such as RCA and HCR allow efficient, rapid analysis and diagnosis. G-quadruplexes coupled with probes can also be used to analyze tumor-related proteins, such as apurinic/apyrimidinic endonuclease 1 (APE1) and carcinoembryonic antigen (CEA), with the RCA or HCR reaction as signal amplification [99,100]. However, the simultaneous detection of two biomarkers remains challenging. By using an aptamer to capture the target protein, it is possible to detect both alpha-fetoprotein (AFP) and glypican-3 (GPC3) in clinical specimens of hepatocellular carcinoma [101], also illustrated in Figure 4a. This design used N-methyl mesoporphyrin IX (NMM) and quantum dots (QDs) as signal reporters and simple enzyme-free parallel CHA as an amplification strategy. After selective binding of the double aptamer dsDNA probe to the target, the released ssDNA triggered CHA amplification: the simultaneous release of G-quadruplex sequence and Ag^+^, which NMM and CdTe QDs selectively recognized, respectively. By fluorescence values, the LOD was as low as 3 fg/mL for AFP and 0.25 fg/mL for GPC3. Leveraging the reverse cleavage energy of the CRISPR-Cas system can also be used to co-detect multiple proteins in exosomes. A novel DNA zyme walker-triggered CRISPR-Cas12a/Cas13a strategy was proposed for the simultaneous detection of serum amyloid a-1 protein (SAA1) and coagulation factor V (FV) [102]. Anti-SAA1 capture antibody (Ab1SAA1) and anti-FV capture antibody (Ab1FV) were immobilized on the surface of carboxylated magnetic beads (MBs1) to form immunomagnetic beads (MBs1@Ab1). The anti-SAA1 detection antibody (Ab2SAA1) and the anti-FV detection antibody (Ab2FV) were coupled to two DNAzyme walkers (W1 and W2) that could bind to the captured SAA1 and FV to form a sandwich complex. The MBs2 track, obtained by coupling two tracks (T1 and T2) to streptavidin-modified magnetic beads (MBs2), was then added to the sandwich complex. In the presence of the coenzyme Mg^2+^, DNAzyme walkers could specifically cleave the tracks and release multiple P1 (ssRNA) and P2 (ssDNA). P1 and P2 hybridized with crRNA and activated Cas13a and Cas12a, leading to cleavage of the reporter genes (ssRNA FQ1 and ssDNA FQ2) and a significant fluorescence (FL) signal. The localization analysis of protein expression has importance in cellular activities.

Unlike the detection methods in the solution described above, electrochemical detection located at the interface shows high sensitivity and rapid analysis. In 2015, an electrochemical DNA sensor with the spatial potential resistance effect was used directly to detect proteins in whole blood [103]. As exemplified in Figure 4b, when a protein banded to a signaling DNA strand, a spatial site blocking effect occurred, which limited the ability of DNA to hybridize with the complementary strand attached to the surface, resulting in a diminished current signal. Similarly, to enable the detection of the highly specific protein biomarker alpha-methyl acyl-CoA racemase (AMACR) in prostate cancer, a spherical nucleic acid (Y–SNAs) consisting of assisted DNA (A1), AMACR aptamer, and CRISPR-Cas12a DNA activator had been designed [104]. In the presence of AMACR, Y-SNAs as target converters could achieve quantitative activation of CRISPR-Cas12a by outputting a DNA activator that was linearly related to the target. The trans-cleavage activity of CRISPR-Cas12a was exploited to release ferrocene-labeled burst probes (QPs) on the electrode surface to trigger the amplified electrochemiluminescence (ECL) signal.

In addition to using the probe alone to capture proteins, DNA scaffolds such as tetrahedrons or origami provide enhanced selective binding affinity and better stability [105]. For example, by clever design, variations in tetrahedral structure can also be used to characterize the presence or absence of matter. A DNA tetrahedron-based biosensor (DTB) was conceived for imaging and detection of the essential RNA interference (RNAi) protein argonaute2 (Ago2) [106], which contains two fragments: a DNA tetrahedron as a framework and a photo-induced electron transfer (PET) pair as a fluorescent sensor, as shown in Figure 4c. The PET pair consists of DNA/silver nanocluster (AgNC) and G-quadruplex/hemin complexes labeled at both ends of the other DNA strand, respectively. The DNA tetrahedral structure forms a switchable scaffold that can present functional DNA motifs in two different modes depending on the presence/absence of the target protein. Therefore, by measuring DNA/AgNC fluorescence, an Ago2 detection limit of 4.54 nM was obtained. Furthermore, DNA origami acts as a powerful carrier platform that can be fabricated into nanopores, and the change in currents due to the transposition of shapes can be used for biosensing [107]. Instead of integrating averaging techniques, the single-molecule level offers small biomarker concentration detection advantages. For example, Raveendran et al. utilize the aptamer-functionalized nanopores of DNA origami to detect human C-reactive protein (CRP) in clinically relevant fluids [108]. In addition to the characteristic different peak shapes, peak amplitudes and dwell times can also be used to distinguish between occupied and unoccupied carriers, contributing to single-molecule sensing. Single-molecule level detection means multiple substances can be simultaneously detected in complex samples. There, solid nanopores enabled highly multiplexed sensing by using digitally encoded DNA nanostructures [109]. A library of DNA nanostructures had been constructed to detect a single, specific antibody, and each member contained a unique barcode. By electrophoretically driving the DNA structure through the solid nanopore, a 3-digit barcode could be assigned with 94% accuracy. The method can simultaneously detect four antibodies of the same isotype at nanomolar concentration levels. Long, linear DNA strands are commonly used for direct antibody barcoding, but these complexes have poor intracellular stability and function. Therefore, 3D barcoding technology has been developed to express and distribute subcellular proteins. Through the interaction of targeting protein-targeting antibodies, a 3D barcoding method that exploited the combined probe sequence and configurational programmability of the DNA tetrahedron was designed [110]. The technique enabled the accurate classification of molecular subtypes of breast cancer and the identification of subcellular spatial markers of disease aggressiveness.

Additionally, in non-solutions, the formation of sandwich complexes with tetrahedra as a scaffold is often used to detect various disease-associated proteins. For example, based on the DNA tetrahedral (NTH)-linked double aptamer and magnetic metal-organic framework, an ultra-sensitive volumetric sensor was constructed for the analysis of cardiac troponin I (cTnI) [111]. In the presence of cTnI, an aptamer-protein-nanoprobe sandwich-type structure was formed that catalyzed the oxidation of hydroquinone by hydrogen peroxide to amplify the electrochemical signal, with a linearly increasing volumetric signal over the concentration range of 0.01–100 ng/mL^−1^ cTnI and a detection limit of 5.7 pg/mL^−1^. Except for the DNAzyme activity of G-quadruplexes as the ECL signal, ECL-RET can be used as an electrochemiluminescence radiometric biosensor [112], as shown in Figure 4d. The tetrahedral DNA probe was immobilized on gold nanoparticles loaded with graphite-phase carbon nitride (Au-g-C3N4). Based on the ratio of ECL intensity at 595 and 460 nm, which improved the binding efficiency of transcription factors and the change in ECL ratio by forming a DNA probe-antigen-antibody sandwich structure, the NF-κB p50 assay was achieved with a detection limit of 5.8 pM and high stability and specificity. Furthermore, by introducing the PDMS microfluidic channels, based on a microfluidic chip integrated with a liquid automatic conveying unit and electrochemical detection platform, a μFEC detection system was realized to detect prostate-specific antigen (PSA) [113]. The actual serum samples can achieve a linear dynamic range of 1–100 ng/mL, a detection limit of 0.2 ng/mL, and a total reaction time of <25 min.

**Figure 4 sensors-23-03313-f004:**
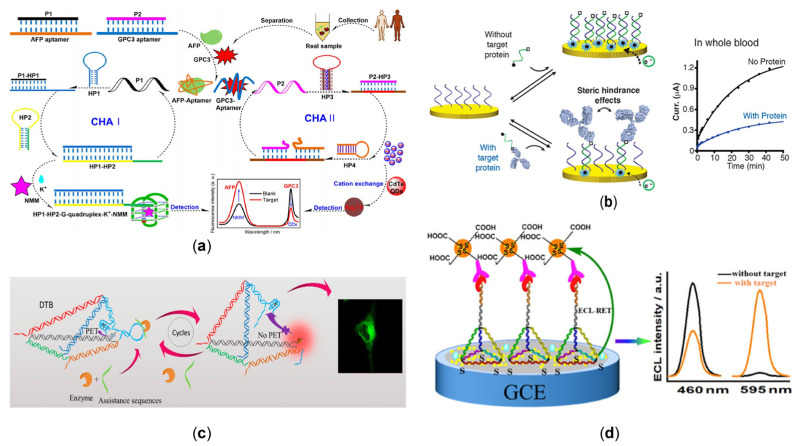
(**a**) Dual CHA system, with the released individual aptamer for the simultaneous analysis of AFP and GPC3 [101]. (Reprinted from *ACS Applied Materials and Interfaces*, Chen, P.; Jiang, P.; Lin, Q.; Zeng, X.; Liu, T.; Li, M.; Yuan, Y.; Song, S.; Zhang, J.; Huang, J.; et al. Simultaneous Homogeneous Fluorescence Detection of AFP and GPC3 in Hepatocellular Carcinoma Clinical Samples Assisted by Enzyme-Free Catalytic Hairpin Assembly, **2022**, *14*, 28697–28705, Copyright (2022), with permission from the American Chemical Society). (**b**) Due to the steric hindrance effects, a highly selective electrochemical sensor based on DNA to detect the presence of macromolecules [103]. (Reprinted from the *Journal of the American Chemical Society*, Mahshid, S.S.; Camiré, S.; Ricci, F.; Vallée-Bélisle, A. A Highly Selective Electrochemical DNA-Based Sensor That Employs Steric Hindrance Effects to Detect Proteins Directly in Whole Blood, **2015**, *137*, 15596–15599, Copyright (2015), with permission from the American Chemical Society). (**c**) The DNA tetrahedral structure forms a switchable scaffold. The cleavage reaction of the Ago2/miR-21 complex opens the hairpin structure, causing the PET pairs to separate in the spatial state [106]. (Reprinted from *Analytical Chemistry*, Zhang, K.; Huang, W.; Huang, Y.; Li, H.; Wang, K.; Zhu, X.; Xie, M. DNA Tetrahedron Based Biosensor for Argonaute2 Assay in Single Cells and Human Immunodeficiency Virus Type-1 Related Ribonuclease H Detection in Vitro, **2019**, *91*, 7086–7096, Copyright (2019), with permission from the American Chemical Society). (**d**) Ratio-type ECL immunosensor for transcription factor (NF-κB p50) determination [112]. (Reprinted from *ACS Applied Materials and Interfaces*, Fan, Z.; Lin, Z.; Wang, Z.; Wang, J.; Xie, M.; Zhao, J.; Zhang, K.; Huang, W. Dual-Wavelength Electrochemiluminescence Ratiometric Biosensor for NF-kappaB p50 Detection with Dimethylthiodiaminoterephthalate Fluorophore and Self-Assembled DNA Tetrahedron Nanostructures Probe, **2020**, *12*, 11409–11418, Copyright (2020), with permission from the American Chemical Society).

### 3.4. Applications of DNA Nanotechnology in Bacteria

#### 3.4.1. Gram-Positive Bacteria Detection

The identification and detection of bacteria are of great research value and have practical significance in disease prevention and control, clinical diagnosis, and food hygiene safety. Gram-positive and Gram-negative bacteria are distinguished by the results of Gram staining of the bacteria. Any stained bacteria with purple bodies are Gram-positive. Many different detection methods have been developed to detect bacteria. A class of fluorophores represented by FAM, Cy3/Cy5, and Texas red can emit from the UV to the NIR region. The interaction between the fluorophore-labeled DNA probe and the target is observed by monitoring the change in fluorescence signal. As shown in Figure 5a, an aptamer-based Cas14a1 biosensor (ACasB) is developed to detect live Staphylococcus aureus (S. aureus) with high specificity and sensitivity [114]. Upon addition of live S. aureus, the blocker can be released upon bacterial binding to the aptamer. The released blocker then forms a ternary complex with Cas14a1/sgRNA, which cleaves the fluorescent reporter from the quencher ssDNA (FQ), resulting in a solid fluorescent signal. The method does not require nucleic acid extraction and amplification and may expand the use of CRISPR-Cas systems in live cell detection. A particle counter is an accurate, fast, and cost-effective instrument that can analyze particles’ size and quantity distribution by recording the frequency and waveform of the corresponding pulse signal [115], which can circumvent some restrictions involving biological enzymes. Ren et al. proposed a method for detecting Lactobacillus monocytogenes DNA by enzyme-free dual signal amplification based on microporous resistance technology combined with polystyrene (PS) microsphere polymerization constructed by CHA [116]. As shown in Figure 5b, A detection probe (probe2) and a trigger DNA (tDNA) are modified on PS microspheres. When target DNA is present, the signal is amplified with the help of a capture probe (probe1), which, together with probe2, forms a complex that triggers the CHA reaction. The microporous resistance technique allows sensitive identification of PS microsphere aggregation caused by the CHA reaction and quantitative analysis of the target DNA.

Bacterial bloodstream infections have high morbidity and mortality, and the ultra-multiplex detection of pathogenic bacteria is increasingly essential. DNA walker was fully considered due to its good performance in human health-related bacteria. An ultra-high throughput bacterial detection system using random DNA walkers in droplets is reported, developing a one-step, rapid, ultra-multiplex detection method [117]. After the random DNA walker has been effectively activated, the signal in the droplet is amplified by mixing the encapsulated bacteria. A precisely controlled ratio and quantity lead to a digital and independent adjustment of color and intensity. The generation of a fluorescent signal makes direct optical readout possible. By combining this with flow cytometry, the strategy shows excellent potential in a wide range of bioanalyses. In addition, the detection method for bacterial metabolites is a relatively novel idea. For example, artificial photosynthesis can enable non-photosynthetic bacteria to generate photogenerated electrons for carbon dioxide fixation. Yang’s team penetrated gold nanoclusters (AuNC) as photosensitizers into the bacterium Moorella thermoacetica, which can directly generate electrons in the bacteria to convert carbon dioxide into acetic acid [118].

#### 3.4.2. Gram-Negative Bacteria Detection

Gram-negative bacteria refer to bacteria with a red Gram staining reaction. Common Gram-negative bacteria, including Escherichia coli, typhoid bacilli, and dysentery bacilli, are one of the primary pathogens causing human infectious diseases. Nanozymes are nanomaterials with enzyme-like properties. Some enzyme-like nanomaterials, such as gold [Au] and platinum [Pt], have been shown to catalyze the oxidation of peroxidase substrates to change the color of the solution, making it possible to achieve biosensing through color change [119,120,121]. Based on the G-quadruplex (G4) DNAzyme as the signal amplifying element, a simple and sensitive colorimetric aptamer sensor was employed for the detection of the widespread food-borne pathogen Vibrio parahemolyticus (V. parahemolyticus), and the detection limit could be as low as 10 CFU/mL [122]. In addition, an aptamer-initiated DNA walker can be used to detect *E. coils* [123]. As shown in Figure 5c, in the presence of the target bacteria, the aptamer (black strand) bonded to the bacteria and simultaneously released two walking strands (red and blue strands). Exo III acted as a power source by consuming thiol-labeled oligonucleotides, and the released walking strands traveled autonomously on an AuNP-based 3D track. The change-inducing color from red to blue could be used as an analytical signal for quantitative analysis of bacteria due to multiple cycles of hybridization and digestion leading to unstable probe aggregation. Then, RNA-cleaving DNAzymes (RCDs) integrating biorecognition with signal transduction were also used to discover *E. coli* [124]. There were capture channels and release channels on the e-differential chip. Changes in the position of the methylene blue led to changes in the current in both channels, combined with intelligent devices for fast and sensitive analysis. The system allowed the development of a fully integrated POC test kit to identify multiple pathogens for rapid, multiplexed, and reagent-free testing. Furthermore, it is not difficult to detect two bacteria simultaneously. A dual-mode sensor was designed to detect Salmonella typhimurium (S.T.) and Vibrio parahaemolyticus (V.P.) with a microfluidic chip. To obtain multiple signals, a series of magnetic DNA-encoded probes (MDEs) containing RCA-DNA rich in G-tetrameric sequences were prepared [125], as shown in Figure 5d. They can bind to hemin as a DNAzyme and catalyze the color development of the TMB-H_2_O_2_ system. This dual-mode aptamer sensor enables rapid in situ visual screening and simultaneous quantification of a wide range of bacteria.

**Figure 5 sensors-23-03313-f005:**
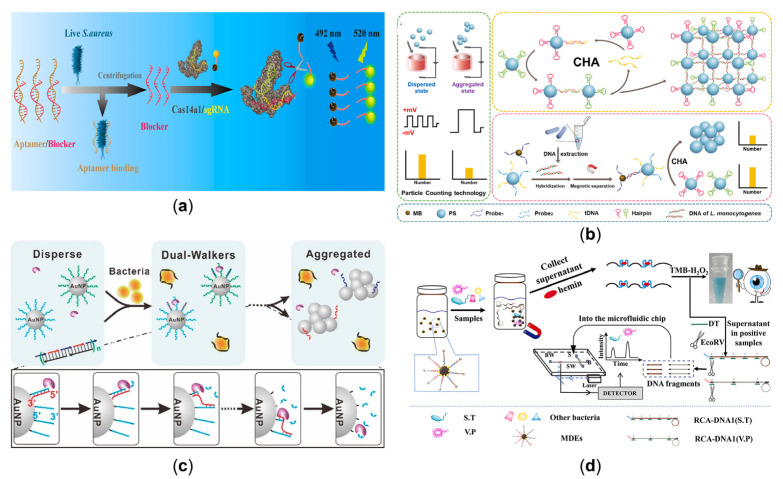
(**a**) The released blocker from the dsDNA triggered by the live S. aureus, combined with Cas14a1/sgRNA to cleave the FQ probe and produce an intense fluorescent signal [114]. (Reprinted from *Biosensors and Bioelectronics,* Wei, Y.; Tao, Z.; Wan, L.; Zong, C.; Wu, J.; Tan, X.; Wang, B.; Guo, Z.; Zhang, L.; Yuan, H.; et al. Aptamer-based Cas14a1 biosensor for amplification-free live pathogenic detection, **2022**, *211*, 114282, Copyright (2022), with permission from Elsevier). (**b**) The PS microspheres in different states based on the CHA-mediated micro-orifice resistance assay using DNA hybridization reaction to detect L. monocytogenes [116]. (Reprinted from *Biosensors and Bioelectronics*, Ren, L.; Hong, F.; Chen, Y. Enzyme-free catalytic hairpin assembly reaction-mediated micro-orifice resistance assay for the ultrasensitive and low-cost detection of Listeria monocytogenes, **2022**, *214*, 114490, Copyright (2022), with permission from Elsevier) (**c**) A stochastic DNA dual walker with two kinds of released multiple walking strands as a colorimetric biosensor for bacterial detection [123]. (Reprinted from *Analytical Chemistry*, Yang, H.; Xiao, M.; Lai, W.; Wan, Y.; Li, L.; Pei, H. Stochastic DNA Dual-Walkers for Ultrafast Colorimetric Bacteria Detection, **2020**, *92*, 4990–4995, Copyright (2020), with permission from the American Chemical Society) (**d**) A dual-mode aptasensor for multiple bacterial tests either by naked eyes or microfluidic chips [125]. (Reprinted from *Talanta*, Yu, J.; Wu, H.; He, L.; Tan, L.; Jia, Z.; Gan, N. The universal dual-mode aptasensor for simultaneous determination of different bacteria based on naked eyes and microfluidic-chip together with magnetic DNA encoded probes, **2021**, *225*, 122062, Copyright (2021), with permission from Elsevier).

### 3.5. Applications of DNA Nanotechnology in Others

In addition to detecting common biomarkers such as nucleic acids and proteins, DNA nanostructures can be used to see several other substances, such as vesicles and small molecules.

Heavy metals are mainly present in the environment as ions and, because they are not biodegradable, can accumulate at all levels of the food chain through biomagnification, causing biological heavy metal poisoning and environmental degradation. They can enter the body through dermal contact and the respiratory and digestive tracts and accumulate in the body by precipitation, leading to neurological disorders, cancer, genotoxicity, and even death. Therefore, accurate quantification of HMs is essential to protect the ecological environment and maintain human health [126,127,128]. A simple, fast, and intuitive distance-based readout on a paper chip for Hg^2+^ detection strategy was devised under the formation of Hg^2+^-mediated G-tetramer-hemin DNAzymes [129]. For another method for Hg^2+^ detection, combining target-induced strand displacement amplification (SDA) and metal ion-dependent DNAzyme cycle amplification, an electrochemical detection method has been developed [130]. Target Hg^2+^ triggered the autonomous synthesis of Mg^2+^-dependent enzyme sequences after forming stable T-Hg^2+^-T structures. The exported Mg^2+^-dependent enzyme sequences can further hybridize with hairpin-structured substrate sequences containing G-tetramers to form Mg^2+^ -dependent DNAzymes on the electrode surface that catalyzed the cleavage of the hairpin-structured substrate in the presence of Mg^2+^, releasing the enzyme sequences to trigger another cleavage cycle and electrode surface, leaving a G-quadruplex fragment. By incubation with Mg^2+^ and free c-myc probes, the retained G-tetramers can spontaneously self-assemble into long linear G nanowires, form associated hemin/G-tetramer repeat units with hemin, and act as signal output devices. In addition, tetrahedra can also be used to detect metal ions [131], as illustrated in Figure 6a. The DNA tetrahedra were located at the end of the substrate/DNAzyme duplex structure, providing a protective barrier against nuclease attack. In the absence of Pb^2+^, FAM fluorescence was effectively burst. When the Pb^2+^ cofactor is present, the DNAzyme exhibits catalytic activity and cleaves the substrate chain, separating the FAM from BHQ-1 and releasing the fluorescence. This strategy allows quantitative detection of Pb^2+^ below 1 nM without interference from non-target metal ions.

Tumor exosomes enhance tumor migration and invasion by modulating the tumor microenvironment. Therefore, detecting cancer-specific exosomes is essential for early cancer diagnosis [132]. A label-free electrochemical aptamer sensor for exosome-specific detection of gastric cancer has been created [133]. The platform comprised an anti-CD63 antibody-modified gold electrode and a gastric cancer exosome-specific aptamer. The aptamer contains a primer sequence complementary to the G-quadruplex template. Thus, the presence of the target exosome triggered a rolling loop amplification and generated multiple g-quadruplex units, which catalyzed the reduction of H_2_O_2_ and generated an electrochemical signal. Extracellular vesicles are also essential biomarkers of the circulatory system in disease diagnosis and prognosis [134]. A 2D magnetic platform combined with a QD imaging system was developed to improve extracellular vesicle capture performance [135]. As illustrated in Figure 6b, the assay was based on two-dimensional flexible Fe_3_O_4_-MoS_2_-aptamer nanostructures, using peroxidase-coupled cholesterol (HRP-cholesterol) to label the lipid bilayer of extracellular vesicles, followed by the introduction of an H_2_O_2_-sensitive QD imaging system whose fluorescence decreased in the presence of H_2_O_2_. The method enabled the quantification of extracellular vesicle markers by smartphone recognition of fluorescence values. Three-dimensional nanoparticles have a high surface area and space utilization, and the 3D DNA walkers can move on the surface of the nanoparticles, resulting in higher amplification efficiencies and high DNA enrichment capabilities [136]. Here, using target exons as a three-dimensional track for the first time, a self-tracking DNA walker (STDW) exploiting an exosomal glycoprotein for wash-free detection of tumor exons was designed [137], which was enabled by the recognition of a cleaved aptamer initiated by an autonomous pathway using the CHA. The STDW method was highly selective and sensitive, directly detecting tumor Exos in cell culture media and serum.

Ochratoxin A (OTA) is one of the most prevalent and toxic fungal toxins in nature and is a secondary metabolite produced mainly by Aspergillus and Penicillium fungi, with severe carcinogenic, hepatotoxic, and nephrotoxic properties [138]. An easy-to-use dual-mode colorimetric and fluorescent strategy was constructed for the detection of OTA by using G4 structures [139]. The colorimetric mode can eliminate autofluorescence interference from sample matrices, while the fluorescent mode overcomes interference from pigments in sample matrices. An alternative low-background electrochemical biosensor for OTA detection was devised based on DNA tetrahedral surround primers and DNAzyme-activated programmed RCA [140]. The design allows the detection of OTA in the range of 1 pg/mL~10 ng/mL with a minimum concentration limit of 0.773 pg/mL. Simultaneous detection of two toxins has also been achieved by changes in the tetrahedral structure of DNA [141], as shown in Figure 6c. A signal-switched fluorescent sensor was constructed based on self-assembled DNA tetrahedra for the simultaneous and rapid detection of OTA and aflatoxin B1 (AFB1). The distance between the fluorophore and the quencher is adjusted according to the variation of the tetrahedral hairpin structure. The result is a LOD as low as 0.005 ng/mL for OTA and 0.01 ng/mL for AFB1. As shown in Figure 6d, another dual-mode aptadetector with fluorescence and surface-enhanced Raman scattering (FL-SERS) was developed for the sensitive and rapid detection of OTA [142]. Gold nanostars carrying OTA aptamers (Apt-AuNSs) and gold nanospheres functionalizing Cy3-modified complementary DNA (cDNA-AuNPs) were assembled into the aptasensor. This aptasensor showed a low FL signal because of the proximity of Cy3 to AuNSs and a high SERS signal because of the “hot spot” effect generated by the nano-gap between AuNSs and AuNPs. In the presence of OTA, the preferential combination of aptamer and OTA produced a weaker SERS signal. At the same time, the cDNA-AuNPs were released from the hybrid complex, resulting in a recovery of the FL signal. The detection limit of the mechanism was 0.17 ng/mL in the FL model and 1.03 pg/mL in SERS mode. In addition to the hot spot effects, a robust chiral signal produced at the position of the surface plasma resonance peak out of the dipole-dipole interaction among particles can be used as an output signal [143]. A DNA-based chiral biosensor involving the self-assembly of a shell core–gold (Au) satellite nanostructure was designed to detect the OTA [144]. A strong chiral signal was generated by the assembly of core-satellite nanostructures based on OTA-aptamer binding, which exhibited an intense circular dichroism (CD) peak. The presence of different levels of OTA limited the integrity of the core-satellite nanostructure assembly to a certain extent. Accordingly, the chiral strength of the community diminished with increasing OTA concentration, thus allowing quantitative determination of the target. The designed chiral sensor showed good linearity between the CD signal and OTA concentration within the range of 0.1–5 pg/mL with a detection limit as low as 0.037 pg/mL.

**Figure 6 sensors-23-03313-f006:**
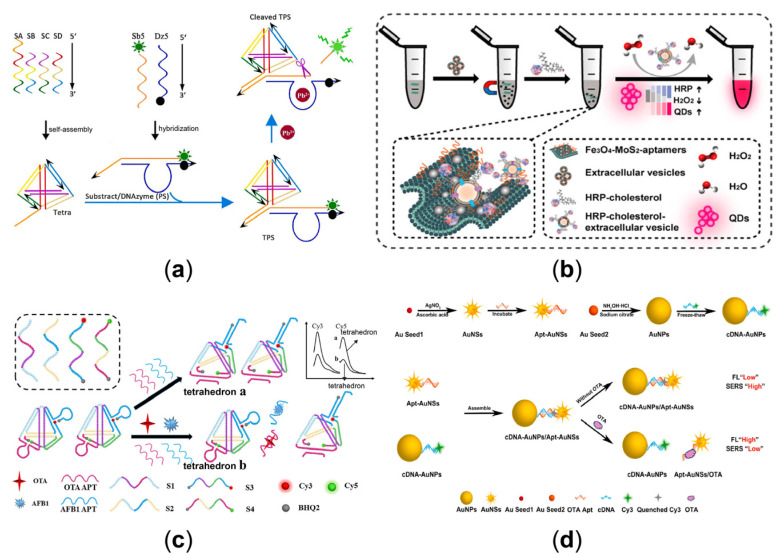
(**a**) Tetrahedron-based Pb^2+^-sensitive DNAzyme sensor (TPS) that combines GR-5 DNAzyme with DNA tetrahedral [131]. (Reprinted from *Talanta*, Guan, H.; Yang, S.; Zheng, C.; Zhu, L.; Sun, S.; Guo, M.; Hu, X.; Huang, X.; Wang, L.; Shen, Z. DNAzyme-based sensing probe protected by DNA tetrahedron from nuclease degradation for the detection of lead ions, **2021**, *233*, 122543, Copyright (2021), with permission from Elsevier). (**b**) The 2D Fe_3_O_4_-MoS_2_ platform for efficient enrichment and test of extracellular vesicles based on peroxidase-coupled cholesterol [135]. (Reprinted from *Analytical Chemistry*, Li, Z.; Ma, D.; Zhang, Y.; Luo, Z.; Weng, L.; Ding, X.; Wang, L. Biomimetic 3D Recognition with 2D Flexible Nanoarchitectures for Ultrasensitive and Visual Extracellular Vesicle Detection, **2022**, *94*, 14794–14800, Copyright (2022), with permission from the American Chemical Society). (**c**) A signal on-off fluorescence sensor based on the self-assembly DNA tetrahedron for the rapid simultaneous detection of ochratoxin A (OTA) and aflatoxin B1 (AFB1) [141]. (Reprinted from *Analytica Chimica Acta*, Ren, W.; Pang, J.; Ma, R.; Liang, X.; Wei, M.; Suo, Z.; He, B.; Liu, Y. A signal on-off fluorescence sensor based on the self-assembly DNA tetrahedron for simultaneous detection of ochratoxin A and aflatoxin B1, **2022**, *1198*, 339566, Copyright (2022), with permission from Elsevier). (**d**) The FL-SERS dual-mode mechanism involving the Apt-AuNSs and the cDNA-AuNPs to detect OTA [142]. (Reprinted from *Biosensors and Bioelectronics*, Wang, H.; Zhao, B.; Ye, Y.; Qi, X.; Zhang, Y.; Xia, X.; Wang, X.; Zhou, N. A fluorescence and surface-enhanced Raman scattering dual-mode aptasensor for rapid and sensitive detection of ochratoxin A, **2022**, *207*, 114164, Copyright (2022), with permission from Elsevier).

## 4. Summary

In the past few years, framework DNA-enabled nanostructures have been extensively researched from structure, function, and application perspectives. Various strategies have been used to develop framework DNA-based biosensors, and successful demonstrations in detecting different biomarkers, including nucleic acids, proteins, viruses, bacteria, etc., have exhibited excellent performances. The superiorities of framework DNA-based biosensors are as follows: first, due to their special molecular structure and physicochemical properties, a nearly infinite number of sequences able to hybridize reliably with their complementary oligonucleotides can be used to fold a variety of 2D, 3D nanostructures with different shapes and functionalities in an orderly fashion. Because of the structural uniformity and integrability nature of DNA nanostructures, framework DNA-based biosensors can be easily and ingeniously accomplished. Second, taking advantage of functional modifications in the conjugation of DNA sequences with biomolecules and ligands, we are now able to expand its couple with varying recognition elements, making F-DNA-based biosensors very universal for detecting many other biomarkers such as exosomes, ions, aflatoxin, and so on [145]. Third, signal amplification approaches can be easily integrated with the aid of dynamic DNA walkers (RCA, HCR, and CHA), preventing tedious steps of operation and temperature cycling in PCR to obtain the detection limit down to the fM level, resulting in sensitive sensing. Finally, based on the excellent flexibility for organizing molecules at the nanoscale, strategies that obtain favorable probe orientation and finely control the precise spatial arrangement of such probes on surfaces provide target accessibility at the heterogeneous biosensing interface for efficient and specific detection.

As novel biosensor components or probes, framework DNA-enabled nanostructures are versatile building modules for sensitive molecular sensing that is unachievable using conventional detecting strategies. These approaches have been advanced from mere proof-of-concept studies to “platform technologies.” Because panels of biomarkers of a specific disease for clinical application have more meaning than a single biomarker, which overlaps with another condition, one trend in this area is the highly sensitive sensing of multiplexed biomarkers. DNA structures with high programmability are natural scaffolds, allowing the detection of a multiplex biomarker or various types of biomarkers in a single assay. Some examples confirmed that DNA nanoswitches were shown to detect five microRNAs in a single assay. In addition, four different microRNAs can be multiplexed to be detected by DNA tetrahedra. Liu et al. used a plasmonic layer coated and S9.6 anti-DNA/RNA duplex antibody functionalized single microbead as a microreactor, an AuNP labeled with both a Raman coding molecule and a DNA probe as a recognition pool for the high-precision multiplexed assay of sub-pM target miRNA in a simple mix-and-read format [146]. Recently, association with the latest techniques and other materials has been another trend in this area. A non-self-fluorescent molecularly imprinted polymeric aptasensor (MIP-aptasensor) was designed to detect the H5N1 virus [147]. Aptamer-functionalized persistent luminescent nanoparticle Zn_2_GeO_4_:Mn^2+^-H5N1 coupled with a magnetic MIP can produce an intense, constant luminescent signal after identifying the H5N1 virus, which has the highly selective and sensitive ability. Combined with the enzymatic resistance, good endocytosis, and non-cytotoxicity, and the predictability and excellent addressability of DNA nanostructures, another trend in this area is intracellular biosensing and imaging of the biological processes for real-time visualization. Wang et al. developed a multi-armed tetrahedral DNA with a cytochrome c (Cyt c) aptamer and telomerase primer for simultaneous fluorescence imaging [148]. The early and late stages (e.g., the whole course) of cell apoptosis were real-time, sensitive, and precisely visualized, showing strong abilities to monitor the oxidative stress of cells. For a better understanding of complex biological systems, F-DNA-based biosensors have been expanded to explore cellular functions, biomolecular interactions, the transport of materials within cells, drug efficacy evaluation, and disease mechanisms.

Concerned with the sensitivity and specificity of sensing approaches, biosensors based on framework DNA are on par with or better than current techniques. To meet the requirements of more comprehensive applications, the area of research will be more oriented toward the practical use of such sensors. Practicality and sensitivity need balance to transition from the laboratory to the real world. Moving forward, a portable reader such as a test paper with DNA-based techniques, which is not feasible to sustain the high sensitivity in the lab but is convenient for anyone, deserves a try. In addition to that, the high cost is also one of the significant challenges. Some studies have integrated aptamers into DNA nanostructures for diagnostics because they do not need chemical modifications for signal generation and detection. Based on this method, DNA tetrahedra combined with multiple aptamers were conceived for quantitatively detecting tumor-associated mucin-1 (MUC-1) protein [149]. In addition to that, probe-functionalized DNA tetrahedra concentrated in the microfluidic interface, which moved the probe away from the surface, and a 3D reaction space was constructed, enabling higher detection efficiency and excellent selectivity in *E. coli* O157: H7 [150]. Multi-biomarkers can be sensed simultaneously by this approach, too. Through ligating different numbers of PtNPs at the endpoints of DNA tetrahedra, signal probes of varying valence states were obtained and used to detect cardiac markers with an adjustable detection range [151]. The aptamer-DNA tetrahedron complex has a synergistic structural advantage that can control the density of such probes and contribute to a renewable probe layer, avoiding nonspecific adsorption of targets on the surface and steric hindrance to the binding of target molecules [152,153,154]. Meanwhile, the intrinsically kinetic and thermodynamic nature of forming these structures has been analyzed to optimize the construction and yield. Real-world biosensing application is also necessary to satisfy large-scale production of delicate frames, especially for macro-materials. Inspired by biotechnology engineering, scientists use intact bacteriophages to produce ssDNA with virtually arbitrary sequences and tailored length for mass production of DNA-origami, a cost-effective and scalable approach [155]. Once addressing this issue, downstream purification after mass assembly is another crucial step for a specific application. In addition to the current means, spin filters, PEG-based separation, magnetic bead capture, and size exclusion columns can be developed for good accessibility of this purpose. These strategies hold potential from the bench to the bedside for clinical applications.

The achievements described can be used in the domain of the integration of diagnosis and treatment. Infectious pathogenic bacteria or viruses threaten the health of millions of people each year, and when they are detected, simultaneous eradication is convenient and important. Zheng et al. built DNA sequence-dependent growth dumbbell-shaped anisotropic gold-platinum nanoparticles with good catalytic and antibacterial activity for the simultaneous detection and eradication of *E. coli* [156]. Under light irradiation, Au-Pt nanoparticles had high photothermal conversion efficiency and enhanced catalytic activity to obtain a detection limit of 2 CFU/mL and efficacious eradication (95%) in 5 min. Bio-mimic broad-spectrum antiviral platforms have been developed for capturing viruses. Based on the programmability and addressability of DNA triangular building blocks, Sigl et al. constructed a self-assembled programmable icosahedral canvas with virus-trapping and anti-viral properties in an icosahedral shell [157], which was durable, commercially available, and easily functionalized. Trapping the virus in a shell to prevent virus-host cell interactions can reduce viral loads in acute infections. Additionally, the research can combine with subject areas such as synthetic biology, polymer material engineering, computer science, and electronics to further explore the physical and chemical essence of framework DNA-based biosensors, which better guide structural architecture and material assembly in a more predictable fashion.

We envision that framework DNA-based materials with more functions and diversity will be valuable platforms for the molecular-level mechanisms of life and provide numerous opportunities for material fabrication, life-like systems, ecological environments, and artificial intelligence.

## Data Availability

Data are available in a publicly accessible repository.

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
