# Peer review of "Programmable Nanostructures Based on Framework-DNA for Applications in Biosensing"

_sensors, 2023, doi:10.3390/s23063313_

Round 1
Reviewer 1 Report
General comment:
In this manuscript, authors tried to review the state of art of nucleic acid nanostructures for biosensing. While it described in detail about the fundamentals of DNA nanotech and a few different directions of application in sensing, authors had missed a major research direction and lots of important literature in the field leading to rapid, low-cost and stable diagnostics. With the suggestion below, a major revision is recommended for authors to integrate the missing parts into the review.
Specific suggestions:
1. G-quadruplexes have the chemical property to catalyze peroxidase activity for signal generation. Authors should review literature on the applications.
2. In addition to just binding, aptamers were often integrated to DNA nanostructure for inducing structural change for signal generation. Authors should also review literature on this topic.
3. There were also lots of papers on applying DNA tetrahedron or other polyhedra for electrochemical sensing or colorimetric assay.
4. Cost is obviously the major issue applying DNA nanostructures for diagnostics. Authors should emphasize on recent research of aptamer-integrated simple DNA nanostructures for diagnostics. Because they do not require any chemical modifications for detection and signal generation.
Author Response
In this manuscript, authors tried to review the state of art of nucleic acid nanostructures for biosensing. While it described in detail about the fundamentals of DNA nanotech and a few different directions of application in sensing, authors had missed a major research direction and lots of important literature in the field leading to rapid, low-cost and stable diagnostics. With the suggestion below, a major revision is recommended for authors to integrate the missing parts into the review.
1. G-quadruplexes have the chemical property to catalyze peroxidase activity for signal generation. Authors should review literature on the applications.
Response:
Thank you for your constructive and positive comments on improving our manuscript. G4-based platforms are ideal candidates for the development of diagnostic sensing probes for DNA, protein, and metal detection owing to cost, thermostability, chemical property and synthesis simplicity. We have supplemented related references in the manuscript in highlights as follows:
The reference [62] is on the page 6, line 257-259.
The reference [63] is on the page 6, line 259-262.
The reference [85] is on the page 9, line 372-379.
The reference [100,101] is on the page 11, line 500-506.
The reference [123] is on the page 15, line 674-677.
The reference [126] is on the page 15, line 691-697.
The reference [130] is on the page 16, line 728-730.
The reference [131] is on the page 16, line 730-742.
The reference [134] is on the page 16, line 751-757.
The reference [140] is on the page 17, line 776-780.
In addition to just binding, aptamers were often integrated to DNA nanostructure for inducing structural change for signal generation. Authors should also review literature on this topic.
Response:
Thank you for your constructive comments. Aptamers integrated to DNA nanostructures which have signal generation by inducing structure change, can effectively improve the capture efficiency of target proteins to increase the sensitivity of the constructed biosensor. Except the literatures cited in our manuscript, we added new literatures as follows with highlights:
The reference [112] is on the page 13, line 581-598.
The reference [141] is on the page 17, line 780-783.
The reference [142] is on the page 17, line 783-789.
The reference [148] is on the page 19, line 870.
The reference [150] is on the page 20, line 893-895.
The reference [151] is on the page 20, line 895-898.
The reference [152] is on the page 20, line 898-901.
The reference [153] is on the page 20, line 901-904.
The reference [154] is on the page 20, line 901-904.
The reference [155] is on the page 20, line 901-904.
There were also lots of papers on applying DNA tetrahedron or other polyhedra for electrochemical sensing or colorimetric assay.
Response:
DNA tetrahedral or polyhedral are discrete three-dimensional DNA nanostructures formed by ingenious design of DNA sequences and the auto-hybridization of multi single strands. They have been widely applied in the fields of microbial identification, medical diagnostics and biosensors due to their good biocompatibility, excellent cell membrane permeability, simple preparation,high yield, and adjustable size and dynamics. We summarize these literatures in highlights as follows:
The reference [66] is on the page 7, line 275-280.
The reference [67] is on the page 7, line 280-284.
The reference [87] is on the page 9, line 381-389.
The reference [107] is on the page 12, line 547-552.
The reference [112] is on the page 13, line 581-598.
The reference [113] is on the page 13, line 588-594.
The reference [114] is on the page 13, line 594-599.
The reference [132] is on the page 16, line 742-748.
The reference [141] is on the page 17, line 780-783.
The reference [142] is on the page 17, line 783-789.
The reference [148] is on the page 19, line 870.
The reference [150] is on the page 20, line 893-895.
The reference [151] is on the page 20, line 895-898.
The reference [152] is on the page 20, line 898-901.
The reference [153] is on the page 20, line 901-904.
The reference [154] is on the page 20, line 901-904.
The reference [155] is on the page 20, line 901-904.
Cost is obviously the major issue applying DNA nanostructures for diagnostics. Authors should emphasize on recent research of aptamer-integrated simple DNA nanostructures for diagnostics. Because they do not require any chemical modifications for detection and signal generation.
Response:
The DNA nanostructure-based sensing platforms still face challenges. One of them is the high cost. Aptamer-integrated simple DNA nanostructures for diagnostics is a good solution because they do not require any chemical modifications for detection and signal generation. We summarize that in line 890-904 in highlights, and the references involved are as followed.
The reference [112] is on the page 13, line 581-598.
The reference [141] is on the page 17, line 780-783.
The reference [142] is on the page 17, line 783-789.
The reference [148] is on the page 19, line 870.
The reference [150] is on the page 20, line 893-895.
The reference [151] is on the page 20, line 895-898.
The reference [152] is on the page 20, line 898-901.
The reference [153] is on the page 20, line 901-904.
The reference [154] is on the page 20, line 901-904.
The reference [155] is on the page 20, line 901-904.
Reviewer 2 Report
Overview of biosensors based on DNA scaffolds. The authors analyzed more than 120 papers on the detection of biomarkers. The article consists of three chapters: introduction, frameworks construction methods and the largest one - its use in biosensors. The authors divide the last chapter of the manuscript into five blocks devoted to various biological objects to be detected. However, the systematization of the collected data is poor. Each of the five blocks considers very different approaches. This makes the review mosaic. Moreover, the authors did not consider all approaches to biosensing presented in the literature.
The title of the manuscript should have a narrower meaning. I would recommend replacing "nucleic acids" with "DNA" since the RNA frameworks was not analyzed.
In addition, the structure of the manuscript looks at least strange. For example, chapters "3.2.1. RNA sequence detection" and following "3.2.2. Aptamer-based detection". This is a different category for division. Next, chapters are "3.3.1. Biomarker Assays" and "3.3.2. Antibody detection". Isn't the antibody a biomarkers?
Authors should better systematize their literature review. Moreover, it is necessary to make a conceptual conclusion about what is currently available, what trends this area in recent years, and where this area of research is going to. And, if it possible possible, try to make your own forecast of in what new areas of scientific research and development the achievements described in the review can be used.
Author Response
Referee #2 (Remarks to the Author):
Comments to the Author
- Overview of biosensors based on DNA scaffolds. The authors analyzed more than 120 papers on the detection of biomarkers. The article consists of three chapters: introduction, frameworks construction methods and the largest one - its use in biosensors. The authors divide the last chapter of the manuscript into five blocks devoted to various biological objects to be detected. However, the systematization of the collected data is poor. Each of the five blocks considers very different approaches. This makes the review mosaic. Moreover, the authors did not consider all approaches to biosensing presented in the literature.
Response:
Thank you for your constructive and positive comments on improving our manuscript. When I was interested in the detection of miRNA, it took a long time to survey many literatures and reviews. Since that, I want to collect and review these examples to satisfy the readers such as me. We wrote this review and divided the last chapter of the manuscript into five blocks. This classification mode directs the readers a well-defined and concentrative discuss on every kind of analyte. In other reviews, if they are interested in one kind analyte such as virus, they need to collect the discrete examples. We have explained the reason in line 84-88.
We have systematized the review, added appropriate literatures and adjusted the location of most researches to make this review easy to understand and not mosaic. The approaches in each of the five blocks are changed to uniform. The blocks of nucleic acids, virus, bacteria and others are classified by the chemical essence of analytes. The block of nucleic acids was divided into “DNA, MicroRNA, mRNA and circle RNA”. The block of virus was divided into “virus nucleic acid and virus protein”. The block of bacteria was adjusted and divided into “Gram-positive bacteria and Gram- negative bacteria”. The detection of ions was added and the block of others was added divided into “ion, exosome and ochratoxin A”. The examples in separate chapters are from simple to complex. The details are in line 235-830.
- The title of the manuscript should have a narrower meaning. I would recommend replacing "nucleic acids" with "DNA" since the RNA frameworks was not analyzed.
Response:
We have replaced "nucleic acids" with "DNA". The “framework nucleic acid” is now changed to “framework DNA” and the abbreviation is “F-DNA”.
- In addition, the structure of the manuscript looks at least strange. For example, chapters "3.2.1. RNA sequence detection" and following "3.2.2. Aptamer-based detection". This is a different category for division. Next, chapters are "3.3.1. Biomarker Assays" and "3.3.2. Antibody detection". Isn't the antibody a biomarker?
Response:
The block of "3.2.1. RNA sequence detection" is based on the nucleic acids of virus and "3.2.2. Aptamer-based detection" is based on proteins of virus, so we divided this part to two categories: “Virus nucleic acid detection” and “Virus protein detection”. Antibody is indeed one kind of biomarkers, so we do not make category in the chapter of protein detection.
- Authors should better systematize their literature review. Moreover, it is necessary to make a conceptual conclusion about what is currently available, what trends this area in recent years, and where this area of research is going to. And, if it is possible, try to make your own forecast of in what new areas of scientific research and development the achievements described in the review can be used.
Response:
We have systematized the review, added appropriate literatures and adjusted the location of most researches to make this review easy to understand and logical. The blocks of nucleic acids, virus, bacteria and others are classified by the chemical essence of analytes. The block of nucleic acids was divided into “DNA, MicroRNA, mRNA and circle RNA”. The block of bacteria was divided into “Gram-positive bacteria and Gram- negative bacteria”. The block of others was divided into “ion, exosome and Ochratoxin A”. The examples in separate chapters are from simple to complex. For example, in the chapter of virus, we divided it into two parts: virus’ nucleic acids and virus’ proteins. In the second part of virus’ proteins, we first introduce a triangular prism DNA (TPDNA) nanostructure as scaffold to detect the virus. Then a well-designed star-shaped DNA architecture with precise spatial pattern-recognition properties that can improve the detection of dengue virus is followed. Similarly, a net-shaped DNA nanostructure for selective recognition and high-affinity capture of intact SARS-CoV-2 through spatial pattern-matching and multivalent interactions was cited. At last, we outlook that some researches will specifically identify the novel mutated strains of SARS-CoV-2. The details are in line 235-830.
We have conceptually concluded about what is currently available in line 832-853. Based on the excellent programmability, structural uniformity, addressability, integrability, easy to modification, enzymatic resistance, good endocytosis and non-cytotoxicity, framework DNA-enabled nanostructures have exhibited excellent performances in detecting different biomarkers, including nucleic acids, proteins, virus, bacteria, ions, exosomes, ions, aflatoxin with high sensitivity and specificity. Its intracellular biosensing and bio-imaging applications have also been demonstrated.
These approaches have been advanced from mere proof-of-concept studies to “platform technologies”. One trend of this area in recent years is multiplexed sensing biomarkers, another is the association with latest techniques and other materials. The third trend is to monitor the cellular functions. The trends have been summarized in line 854-883.
To meet the requirements of wider applications, the area of research is going to more oriented toward practical use of such sensors. For the transformation from laboratory setting to the real-world, there is a long way to go. We have proposed some issues and the solutions in line 884-915.
In my opinion, there are two directions of scientific research and development the achievements described in the review can be used. One direction is the domain of integration of diagnosis and treatment. The other is combination with subject areas to further explore the physical and chemical essence of Framework DNA-based biosensors in a more predictable fashion. The details are in line 916-933.
Round 2
Reviewer 1 Report
Thank you for the revision from authors! Authors have clearly addressed the missing angles in this review and cited relevant references. However, there are lots of writing issues and grammatical mistakes in the manuscript. Authors are recommended to review their writing carefully before publishing.
Author Response
Thank you for the revision from authors! Authors have clearly addressed the missing angles in this review and cited relevant references. However, there are lots of writing issues and grammatical mistakes in the manuscript. Authors are recommended to review their writing carefully before publishing.
Response:
Thank you for your constructive and positive comments on improving our manuscript. We have reviewed the writing, corrected writing issues and grammatical mistakes in the highlight texts.
Reviewer 2 Report
The review now is more structured. The discussion section is interesting and suppose the prospects for the evolution of F-DNA biosensors.
The paper can be published after minor revision. Mostly typos need to be corrected (for example, lines 246,268, 276; what does mean abbreviation RCA; in the line 917 authors wrote "In my opinion..", etc.).
Author Response
The review now is more structured. The discussion section is interesting and suppose the prospects for the evolution of F-DNA biosensors.
The paper can be published after minor revision. Mostly typos need to be corrected (for example, lines 246,268, 276; what does mean abbreviation RCA; in the line 917 authors wrote "In my opinion..", etc.).
Response:
Thank you for your constructive and positive comments on improving our manuscript. RCA is the abbreviation of rolling circle amplification. We have defined this in line 324. "In my opinion.." in the line 917 is inappropriate, we have deleted it. We have corrected the typos in the highlight texts.